starMC: an automata based CTL* model checker

Amparore Elvio Gilberto
Donatelli Susanna susanna.donatelli@unito.it
Gallà Francesco
Dipartimento di Informatica, Università degli Studi di Torino , Torino , Italy
Mousavi Mohammad Reza
Electronic publication date: 2022 Feb 24
Publication date: 2022
Volume: 8
Electronic Location ID: e823
Received 2021 Aug 27; Accepted 2021 Nov 29
Copyright: © 2022 Amparore et al.
Copyright year: 2022
Copyright holder: Amparore et al.
License: This is an open access article distributed under the terms of the Creative Commons Attribution License, which permits unrestricted use, distribution, reproduction and adaptation in any medium and for any purpose provided that it is properly attributed. For attribution, the original author(s), title, publication source (PeerJ Computer Science) and either DOI or URL of the article must be cited.
License URL: https://creativecommons.org/licenses/by/4.0/

Keywords: Model-checking, System verification, Petri nets, CTL* logic, Tools, Buchi automata

Funding: The authors received no funding for this work.

==============================
Model-checking of temporal logic formulae is a widely used technique for the verification of systems. CTL ∗ is a temporal logic that allows to consider an intermix of both branching behaviours (like in CTL) and linear behaviours (LTL), overcoming the limitations of LTL (that cannot express “possibility”) and CTL (cannot fully express fairness). Nevertheless CTL ∗ model-checkers are uncommon. This paper presents (1) the algorithms for a fully symbolic automata-based approach for CTL ∗, and (2) their implementation in the open-source tool starMC, a CTL ∗ model checker for systems specified as Petri nets. Testing has been conducted on thousands of formulas over almost a hundred models. The experiments show that the fully symbolic automata-based approach of starMC can compute the set of states that satisfy a CTL ∗ formula for very large models (non trivial formulas for state spaces larger than 10480 states are evaluated in less than a minute).

Introduction

Temporal logics like LTL (Pnueli, 1977), for linear behaviour, and CTL (Clarke & Emerson, 1981), for branching behaviour, have been successfully used to specify sequential and concurrent systems, and have been widely adopted in many industrial contexts: see, e.g. (Eisner & Fisman, 2016) or the success stories of the model checkers SPIN (Holzmann, 2004) and nuSMV (Cavada et al., 2014). Although LTL and CTL model-checking procedures are known to be quite expensive, since they incur in the so-called “state-space explosion” problem, the use of techniques based on decision diagrams allows, in many cases, to solve industrial-size systems (Burch et al., 1992). In this paper we use “symbolic” to refer to any technique based on some form of decision diagram.

CTL ∗ (Emerson & Halpern, 1983, 1986) is a temporal logic that allows to express linear and branching behaviours in a single property. CTL ∗ allows to go beyond the linear/branching dichotomy (or LTL/CTL), and it is considered a powerful specification language for discrete events dynamic systems. CTL ∗ is known to be strictly more expressive than CTL and LTL. Rozier, in her 2011 survey (Rozier, 2011). Considers CTL ∗ as an adequate logic to overcome LTL and CTL limitations, and in that survey she observes that “the lack of industrial model-checking tools that accept CTL ∗ specifications is a deterrent to the use of this logic” (Rozier, 2011, p.175, end of Sec. 3.2.3), and indeed the situation has not changed much since then.

CTL ∗ properties are of practical interest, and CTL ∗ model checking has been extensively studied in the past. However, as pointed out in Amparore, Donatelli & Gallà (2020b), very few CTL ∗ model checkers exist today, despite its usefulness in specifying both recurrent behaviours and possibility in the same property (something happening infinitely often and for all behaviour, a certain event/state is reachable), or to explicitly express fairness constraints. LTL can express both fairness constraints and recurrent behaviours, but this is not possible in CTL. Instead, CTL can express possibility, which is not possible in LTL.

There are different ways of verifying CTL ∗ properties: with an ad-hoc model-checker based on the identification and verification of LTL sub-formulae (Emerson & Lei, 1987); or through a translation from CTL ∗ to μ-calculus (Kozen, 1983), as defined in Dam (1990, 1994) and revised in Cranen, Groote & Reniers (2011). μ-calculus is indeed known to subsume CTL ∗, but translated formula may be unintuitive to understand, and the translation could lead to an exponential growth of the expression terms.

The Petri net (or PN for short) is a formalism well-suited for the description and the analysis of Discrete Events Dynamic System (DEDS). They have been successfully used in a large variety of application fields, including industrial production (Zurawski & Zhou, 1995; van der Aalst, 1994), business workflow (Aalst, 1998), digital circuits synthesis and analysis (Kondratyev et al., 1998) and system biology (Koch, 2019). This success is mainly due to the particular mixture of ease of specification and good support for the analysis. Analysis techniques can prove properties using only the structure of the PN, or its state space (which is typically exponential in the size of the structure and of its initial state).

The model checking of temporal properties for Petri nets is supported by numerous tools (Kordon et al., 2019). In the last decade the model checking of PN has seen a boost in interest, possibly motivated by the lively Model Checking Competition (MCC) (Kordon et al., 2019) and advances in DD-based implementation of state space exploration based on saturation (Ciardo, Lüttgen & Siminiceanu, 2001).

The contribution of this paper is to clearly identify the algorithms and the steps of a fully symbolic procedure for computing the sat-set of a CTL ∗ formula (the set of states of a given model that satisfy the formula). The implementation of such a procedure for systems specified as Petri nets results in starMC, a tool that computes the sat-set of a CTL ∗ formula using a Büchi-based model checking algorithm. Given a CTL ∗ formula, the algorithm identifies the LTL sub-formulae of maximal length and uses the Spot library (Duret-Lutz, 2014; Duret-Lutz et al., 2016) to translate each of them into a Büchi automaton. The sat-set of each LTL sub-formula is then built from the sat-set of the EfairG(true) on the synchronized product of the model state space and the generated automaton. States, transitions among states, and the synchronized products are all encoded as Decision Diagrams, using the Meddly (Babar & Miner, 2010) library. starMC can be run as a stand-alone tool or as part of the GreatSPN (Amparore et al., 2016) tool suite through the built-in graphical interface (Amparore, 2014).

We could only find another Petri net tool that can deal directly with CTL ∗: LTSmin (Kant et al., 2015). This tool translates CTL ∗ into μ-calculus, using the procedures defined in Dam (1990). The search for a CTL ∗ model-checker was not more successful when considering input languages other than Petri Nets.

Validation of starMC has been achieved taking advantage of the MCC benchmark (Kordon et al., 2019) that comprises models, LTL and CTL formulae, and associated truth values for the models’ initial state.

The benchmark has been enriched with CTL ∗ formulae, derived from the available CTL ones by randomly omitting path quantifiers. Sat-sets of the LTL, CTL, and CTL ∗ formulae have been compared against the sat-sets obtained from the CTL ∗ module of LTSmin. Sat-sets of CTL formulae have also been checked against the CTL model checker in the distribution of GreatSPN.

A motivation for building starMC was teaching and training: to provide students with a uniform environment in which to reason on, and to experiment with, LTL, CTL and CTL ∗ specifications. This motivation follows an earlier re-shaping of GreatSPN to support teaching (Amparore & Donatelli, 2018b). But the main motivation was to investigate a number of open research questions: R1: it is possible to realize an efficient and fully symbolic implementation of CTL ∗ based on Büchi automata that can check models of industrial interest? R2: the approach of R1 is any better than a translation to μ-calculus? R3: variable ordering techniques that exploits the PN structure, like the ones developed in Amparore et al. (2019) for state-space exploration, can be successfully applied to CTL ∗ model-checking? R4: can a Büchi-based approach favour the formulation of counterexamples and witnesses? This paper addresses R1 and R2, and uses the techniques presented in Amparore et al. (2019) for variable ordering. An extensive comparison of different variable order heuristics, as well as the counter-examples generation of R4, are left for future work.

The starMC tool basic principles and structure were first presented in a (demo) paper (Amparore, Donatelli & Gallà, 2020b): here we take a deeper, and at the same time broader, view on CTL ∗ model-checking and on its implementation in starMC. We provide distinct logical and implementation views of the model-checker, thus separating the algorithms from their fully symbolic implementation. The two views and the chosen level of detail, together with a fully open-source code, are meant to ease the scientific and implementation work needed to extend other verification engines with a CTL ∗ model-checker, or to add to starMC additional techniques, for example for optimization or parallelization. We have also added a deeper investigation on previous work to precisely link the approach used with their original sources of inspiration. This paper also introduces a user view-point on starMC, for those interested only in using the tool. Finally, the tool validation has been improved, including less stringent execution time limits, more models and properties, and a deeper investigation of the obtained results.

The paper is organized as follows: “Background” introduces basic definitions and background, “starMC: the outer view” provides an overview of starMC from an user view-point, “starMC: the logical view” describes, at a logical level, the CTL ∗ model-checking procedure, followed by, in “Previous Work”, a discussion and comparison with previous work. The DD-based implementation of the algorithms of “starMC: the logical view” is discussed in “starMC: The Inner View”. “Testing Results” describes the tests that have been conducted, and “Conclusion” concludes the paper.

Background

This section briefly recalls the PN formalism and the CTL ∗ logic. It also introduces the main feature of the GreatSPN tool and of the Meddly and Spot libraries.

Petri nets

A place-transition (P/T) Petri net N is defined (Murata, 1989) as a tuple N=⟨P,T,A,W⟩, where P is the set of places, T is the set of transitions, A⊆(P×T)∪(T×P) is the set of arcs, W:A→N≥1 is the arc weight function. A Petri net system is the pair NS=⟨N,minit⟩ where minit:P→N is the initial marking. Markings represent states of the system, i.e. assignments of tokens to places. A transition t ∈ T is enabled if and only if all input places p of t contain at least W(p, t) tokens. The firing of t removes W(p, t) from all input places p and adds W(t, p′) to each output place p′. Notation m→tt′ indicates the firing of t in marking m, which leads to marking m′. A firing sequence σ for marking m is a sequence of transitions t0, t1,… for which it exists a sequence of markings m0, m1,… such that m = m0 and, ∀i≥0:mi→tmi+1. Accordingly, the sequence of markings π = m0, m1,… identifies a path of NS starting in m0, with π[i] = mi. We indicate with π[i··] the suffix of π starting in mi. We define PM∞(m) as the set of all infinite paths starting in m. From now on we shall interchangeably use the terms “marking” and “state”. The reachability set (RS) of a Petri net system NS is the set of all markings reachable from minit, RS(NS)={m|∃s∈T∗minit→σm}. A reachability graph (RG) of a PN is a tuple RG(NS)=⟨RS,E,minit⟩ where: RS is the set of all markings reachable from minit, E={(m,t,m′)∈RS×T×RS|m,m′∈S,t∈T,m→tm′} is the set of state transitions. If there is no risk of confusion, the NS indication is usually dropped and we write RS and RG.

Figure 1A shows a Petri net with 3 places and 4 transitions, the RS built from the initial state minit = m1, in Fig. 1B, and the corresponding RG, in Fig. 1C.

Figure 1 A Petri net, its RS and RG, and the corresponding Kripke model.

A colored Petri net is a P/T net in which tokens have identities, to allow for a parametric and more compact description of systems. Each colored Petri net can be unfolded into a P/T net of isomorphic RG.

Decision diagrams for state space representation

Decision Diagrams (DD) are a well-known data structure to efficiently encode functions as well as large sets of structured data through their characteristic function. A Binary DD (BDD) encodes a boolean function and a Multivalued DD (MDD) encodes an integer function. A MDD can be used to encode a RS by associating each place of the net to a level in the MDD and ensuring that each path in the MDD corresponds to a reachable state (and vice-versa). Given the example Petri net of Fig. 1A, the MDD of the initial state m1 is depicted in Fig. 2A, and the one of RS in Fig. 2D. DD levels encode the token counts of each Petri net place, and appear in the order P2, P1, P0 (top-down). So the rightmost path in the MDD of RS corresponds to the reachable state [m(P2) = 1, m(P1) = 0, m(P0) = 0], or [1, 0, 0] for short. The depicted MDDs are fully-reduced (when all edges out of a node lead to the same down node, the node is removed and by-passed), so the leftmost path of this MDD encodes both [0, 0, 0] (no token in any place) and [0, 0, 1] (one token in P0).

Figure 2 Decision diagrams for the state space generation of the net in Fig. 1A.

A DD encoding a relation function is called matrix diagrams (M×D). These kind of DDs are used in starMC to represent transitions among states due to the firing of Petri net transitions. An M×D has twice the levels of an MDD, and each pair of levels (also called the unprimed and primed levels for a variable) encodes the before/after relations of a variable (i.e. a place of a Petri net or a location of an automaton). Each of the four M×Ds in Fig. 2B encodes the transformation performed by a single transition of the example net while the M×D in Fig. 2D is the Next-State-Function (NSF), the union of all the single transition M×Ds. The depicted M×D are identity-reduced, i.e. skipping a level pair means that the encoded before-after relation is the identity. The image of this M×D applied to an MDD encoding a state m returns an MDD with all states that can be reached from m by the firing of a single transition. Its fixed point application results in the MDD of the RS. In practice the more efficient saturation technique (Ciardo, Lüttgen & Siminiceanu, 2001), also based on the NSF, is usually preferred. Note that usually the M×D of the RG is never built and stored, as any info in the RG can be retrieved through the DDs of NSF and RS.

In Fig. 2 places are allocated to MDD (and M×D) levels in alphabetical order. It is well known that the DD variable order (the order in which variables are assigned to levels) may greatly influence the size of the resulting DD. The interested reader may find in Amparore, Donatelli & Ciardo (2020a) a recent study on various variable order heuristics that have been successfully applied to encode state spaces of Petri nets.

The temporal logic CTL ∗

CTL ∗ (Emerson & Halpern, 1983, 1986) is a temporal logic in which path operators can be quantified or not, and it therefore allows to freely mix linear and branching reasoning. It subsumes the linear logic LTL (Pnueli, 1977) and the branching logic CTL (Clarke & Emerson, 1981), that are known to have a different expressive power, although with non-empty intersection.

Definition 1 (CTL ∗ syntax). CTL ∗ formulae are the formulae ψ inductively defined by:

Ψ::=a|Ψ∧Ψ|¬Ψ|Eϕ|Aϕ

ϕ::=Ψ|ϕ∧ϕ|¬ϕ|Xϕ|ϕUϕ

where a ∈ AP and AP is a set of atomic propositions. The rules ψ and ϕ define state and path formulae, respectively.

Note that in principle just one type of quantification is needed, as Aϕ = ⫬ E⫬ ϕ. The BNF grammar of CTL ∗ is ambiguous, as boolean expressions appear both as state and path formulae. For instance, the expression a1 ∧ a2, with a1, a2 ∈ AP, is both a state formula (rule ψ ∧ ψ) and a path formula (rule ϕ ∧ ϕ where ϕ ⇝ ψ and ψ ⇝ a).

The satisfaction relation for a CTL ∗ formula is given over a Kripke model M = (S, E, L) with an associated set AP of atomic propositions, where S is a finite and non-empty set of states, E:S→2S is the total successor function, L:S→2AP is a labelling function, and L(s) is the set of atomic propositions that holds in s. The set of paths starting in s, PM∞(s) is defined as for Petri nets. It is straightforward to map a reachability graph into a Kripke model.

Definition 2 (Kripke model of an RG). The Kripke model of a reachability graph RG=⟨RS,E,minit⟩ is the Kripke model M(RG)=⟨RS,E′,L⟩, with E′=E∪{(m,m):∀deadlockstatesm∈RS} being the stuttering of E, and L a labelling function defined over the markings.

Stuttering of deadlock states ensures that the paths in the RG are infinite, as required by a Kripke model. Figure 1D shows the Kripke model derived from the RG in Fig. 1C when considering a set of atomic propositions AP = {β: (#P1 > 0 or #P2 > 0)} and the labelling function depicted next to the states. The two states s3 and s4 are stuttered, since they correspond to the deadlock markings m3 and m4.

Definition 3 (CTL ∗ semantics). The satisfaction relation of CTL ∗ state formulae is defined by:

s⊨aiffa∈L(s)s⊨Ψ1∧Ψ2iff(s⊨Ψ1)and(s⊨Ψ2)s⊨¬Ψiffnots⊨Ψs⊨Aϕiffπ⊨ϕ,forallπ∈PM∞(s)

while the satisfaction relation of path formulae for a path π is:

π⊨Ψiffπ[0]⊨Ψπ⊨ϕ1∧ϕ2iff(π⊨ϕ1)and(π⊨ϕ2)π⊨¬ϕiffnotπ⊨ϕπ⊨Xϕiffπ[1⋯]⊨ϕπ⊨ϕ1Uϕ2iff∃j≥0:(π[j⋯]⊨ϕ2∧(∀0≤k<j:π[k⋯]⊨ϕ1))

A CTL ∗ formula in which all path operators are prefixed by a quantifier is also a CTL formula. To express the relation between CTL ∗ and LTL it is convenient to introduce the following definitions:

Definition 4 (LTL formula). An LTL formula φ is inductively defined by:

φ::=a|φ∧φ|¬φ|Xφ|φUφ

It is usually assumed that path formulae over a Kripke model are “implicitly universally quantified”. To avoid any confusion we assume that the quantifier is explicitly indicated.

Definition 5 (Quantified LTL formula). Given an LTL formula φ a quantified LTL formulae ψ is defined by ψ :: = Eφ | Aφ

The GreatSPN tool

GreatSPN (Amparore et al., 2016; Amparore & Donatelli, 2018b) is a collection of Petri nets tools that are integrated in a common framework, with a unified graphical user interface (Amparore, 2014). Initially developed for performance evaluation of stochastic Petri nets, the tool was enriched over the years with algorithms for qualitative analysis.

From the graphical interface it is possible to draw a Petri net (colored or P/T), to compose Petri nets, to play the token game to familiarize with the net behaviour, and to verify the model using techniques specific to Petri nets (structural properties like P- and T-semiflows) or to check standard Petri net properties (like absence of deadlocks, boundedness and liveness) as well as CTL properties. The state space exploration algorithm and the CTL model-checker are fully symbolic, built using the MDD library Meddly (Babar & Miner, 2010), and a large set of heuristics for variable ordering (Amparore, Donatelli & Ciardo, 2020a) that exploit the net structure. Qualitative verification can be integrated by a stochastic verification, as GreatSPN includes a model-checker (Amparore & Donatelli, 2018a) for the stochastic logic CSLTA (Donatelli, Haddad & Sproston, 2009), and the computation of standard performance indices based on efficient analytical solutions of Markov chains and Markov Regenerative Processes, as well as simulation. The tool is open source and it is available from GitHub (https://github.com/greatspn/SOURCES).

The supporting libraries: meddly and spot

Spot 2.0 (Duret-Lutz et al., 2016) is a framework which provides, among other features, a wide range of tools to manipulate automata over infinite words and to translate them from LTL propositions. The framework has been maintained for decades and can be considered the state-of-the-art of LTL-to-Büchi translation. Our model checker uses the Spot library to construct a Generalized Büchi Automaton with state-based acceptance (SGBA) from each LTL formula encountered during the verification process. We chose to use the Hanoi-Omega Automata (HOA) format (Babiak et al., 2015) to represent such automata. The HOA format is a text-based encoding which supports various derivation of ω-automata. Spot can be downloaded from its home page (https://gitlab.lrde.epita.fr/spotandspot.lrde.epita.fr).

Meddly (Babar & Miner, 2010), Multi-terminal and Edge-valued Decision Diagram LibrarY, is an open-source library (https://github.com/asminer/meddly) for decision diagrams, developed at Iowa State University. It supports the construction and manipulation of various kinds of decision diagrams: Binary and Multivalued decision diagrams, matrix diagrams (M×D) and edge-valued DDs, including a built-in function to support efficient state-space construction based on saturation.

Starmc: the outer view

This section describes how to use the tool for checking CTL ∗ properties. We do so through an example, adapted from (Emerson & Halpern, 1986) and meant to underline the differences between LTL, CTL, and CTL ∗, and the advantage of being able to specify and verify CTL ∗ properties. These differences are sometimes subtle and it is indeed convenient to be able to express properties in the three logics in the same tool (and actually from the same window).

The starMC tool accepts a CTL ∗ syntax (Amparore, Donatelli & Gallà, 2020b) that is more expressive than the minimal one in Definition 1, as it includes more boolean connectors, and the path operators F (in the Future, for some state in the path) and G (globally, for all states in the path). Since CTL ∗ formulae are written in textual mode in the tool, the symbols: &&, ||, !, ,>=, <=, == are used for ∧, ∨, ⫬, ≥, ≤, =, leading to:

Ψ::=AP|Ψ&&Ψ|Ψ ||Ψ|!Ψ|Eϕ|Aϕ

ϕ::=Ψ|ϕ&&ϕ|ϕ ||ϕ|!ϕ|Xϕ|Fϕ|Gϕ|ϕUϕ

AP is the grammar element for atomic propositions, which are formulated over the Petri net elements. AP are defined as boolean expressions over the enabling of transitions and the marking of the places.

AP::=true|false|deadlock|initial|en(T′)|Θ⋈Θ

Θ::=n|#p|bounds(P′)|−Θ|Θ∘Θ

where n∈N, p ∈ P, P′⊆P, T′⊆T, ⋈ ∈ {==, <, <=, >, >=} is a comparison operator, and ∘ ∈ {+, −, ∗, /} is an arithmetic operator. The state formula deadlock evaluates to true for all states that do not enable any transition; initial is true in the initial state of the system; en(T′) is satisfied in all states enabling at least one transition t ∈ T′; #p evaluates to the cardinality of place p in the current marking; bounds(P′) is the maximum sum of token counts of all places in P′ in every reachable marking.

Figure 3 is a screen-shot of the GreatSPN GUI while editing a Petri net model. The model represents the abstract behaviour for a mutual exclusion algorithm in which two processes1 may either stay out of the critical section at will (place NonCSi and self-loop with transition StayNonCSi) or may receive an interrupt (firing of transiton recIRQi) to then proceed to request access to the mutually exclusive resource (place TryCSi).

Figure 3 A screenshot of the GreatSPN interface when drawing the Petri net model for the Mutual Exclusion Problem.

The relevance of this example to CTL ∗ is pointed out very clearly in the work (Emerson & Halpern, 1986) of Emerson and Halpern, that proposes the following verification steps. If we want to specify that the mutual exclusion system should have both behaviours (either process i never tries to access, or at a certain point it will try to access), we may write the quantified LTL formula:

Prop1 : A(G(#NotCSi == 1) || (F #TryCSi == 1))

The model does indeed satisfy the specification of Prop1. Note that the formula is satisfied also if no process is allowed to stay forever in place #NotCSi, given that on all paths it is possible to try to access the critical section. If we want instead to be more specific and add the more stringent requirement that both behaviours should be present (even if on different paths), we can write:

Prop2 : EG(#NotCSi == 1) && EF(#TryCSi == 1) && A(G(#NotCSi == 1) || (F #TryCSi == 1))

Prop2 is not an LTL formula, and no equivalent LTL formula exists, as proved in Emerson & Halpern (1986). Prop2 is not a CTL formula either since the third term has no equivalent CTL. Prop2 is also true, but if we modify the net model to include a singular behaviour for process 1, so that it never receives an interrupt (modelled by removing transition recIRQ1) then for process 2 both properties are true, but for process 1 Prop1 holds while Prop2 does not. So Prop1 (an LTL formula) is not able to discriminate (and therefore to specify) the modified behaviour.

starMC is available both as a command line tool or from inside the GUI. Figure 4 shows a screenshot of the query interface of the GreatSPN GUI when checking the above propositions for the modified mutual exclusion model. The model checker receives a list of queries to be verified, with their language type (LTL, CTL, CTL ∗) explicitly specified. LTL formulae are implicitly quantified as “forall paths”. The queries tagged as STAT and DD provide statistics and a pdf graphical representation of the DD of the RS. The queries tagged CTL are checked with the standard CTL model checker of GreatSPN (Amparore, Beccuti & Donatelli, 2014), while the queries tagged as LTL and CTL ∗ are checked with starMC. Once a query is computed, the GUI shows if it holds in the initial state (true/false), as well as the cardinality of the sat-set (together with the RS size). For the modified mutual exclusion model being examined, | RS| = 3.

Figure 4 A screenshot of the interface that allows to insert queries for the starMC model checker.

Indeed Prop1 is true for both processes, while Prop2 is false for the first process and true for the second one, showing the higher discriminating power of CTL ∗.

Starmc: the logical view

Background: LTL model checking

It is well-known, as in Vardi (1995) or Baier & Katoen (2008, p. 429), that for every LTL formula Aφ it is possible to generate a Generalized Büchi Automaton that accepts all and only the paths that satisfy ϕ, so we shall define them next.

Definition 6 (GBA). A Generalized Büchi Automaton (GBA) is a tuple A=⟨Q,AP,δ,Q0,F⟩, where Q is a finite set of locations, AP is a set of atomic proposition labels, δ⊆Q×AP×Q is a total transition relation, Q0⊆Q is the set of initial locations, and F={Fi|Fi⊆Q}i=1n is the set of accepting sets of locations.

A GBA in which there is a single accepting set ( |F|=1) is called a Büchi automaton (BA tout-court).

The language L( A) of a GBA A is the set of infinite words on AP recognized by a run of A that visits infinitely often at least one location in each Fi∈F.

The standard automata-theoretic approach for model checking LTL formulae over a Kripke model M, following the schema of Vardi & Wolper (1986, 1994) is outlined in Algorithm 1, that checks if the quantified LTL formula Aϕ is satisfied for all paths π starting in the initial state m0. To do so it builds the GBA A of ⫬ϕ (line 3), the BA M of the Kripke model M (line 4), and the synchronized product GBA M⊗A (line 5), whose language is the intersection of the languages of M and A. If this language is empty, no path π satisfies ⫬ϕ and therefore ϕ holds on all paths π∈PM∞(m).

Algorithm 1 Check LTL minit |= Aφ as: ∀π∈PM∞(minit): π|= φ.

1: procedure check∀LTL(M, φ)	
2:  φ′← ⫬φ	
3:   A ← translate φ′ into a Generalized Büchi Automaton	
4:   A ← BA of the Kripke model M(RG)	
5:   M ← synchronized product of A and A	
6:  if ℒ ( M⊗A) is empty then return true	
7:  else return false	
8:  end if	
9: end procedure	

When dealing with Petri nets, the model to be checked is a stuttered RG, which is translated into the Kripke model M( RG) according to Definition 2. The corresponding automaton M is then built (line 4). Note that Büchi automata recognize words over edges, while Kripke models define a language of sequence of states (and associated atomic propositions). The transformation takes as accepting set F={RS} and moves the labelling function of a state to all the incoming arcs, which requires a translation, based on Vardi & Wolper (1986), for the initial state (that has no incoming arc) that consists in adding a fictitious initial state spre, leading to the following definition.

Definition 7 (BA of a Kripke model). The Büchi automaton M=⟨Q,AP,δ,Q0,F⟩ of a Kripke model M = 〈S, E, L〉, with L:S→2AP, for the initial states S0⊆S, is defined as follows: Q = S ∪ {spre}, with spre ∉ S; Q0 = {spre}; ∀s, s′∈ Q\ spre: δ(s, s′, l) ∈ δ iff (s,s′) ∈ E and l = L(s′); ∀s′∈ S0, δ(spre, s′, l) ∈ δ iff l = L(s′); F={S}.

Figure 5A shows the Büchi automaton built for the Kripke model in Fig. 1D, for S0 = s1.

Figure 5 The Büchi automata built by the LTL model checking procedure in Algorithm 1.

We now define a simplified synchronized product of Büchi automata, for the case in which one automaton is a GBA and the other is a BA derived from a Kripke model (therefore with a single acceptance set that includes all states).

Definition 8 (Synchronized product GBA M⊗A). Given a BA M=⟨QM,AP,δM,Q0M,FM={QM}⟩ and GBA A=⟨QA,AP,δA,Q0A,FA⟩, their synchronized product is the GBA M⊗A=⟨Q,AP,δ,Q0,F⟩ where

Q=QM×QA

Q0=Q0M×Q0A

F={Fi}i=1n:∀qA∈FiA,∀qM∈QM:(qA,qM)∈Fi

δ=⊆Q×AP×Q:((qiA,qiM),a,(qjA,qjM))∈δiff

(qiA,a,qjA)∈δAand(qiM,a,qjM)∈δM

It is known (e.g. Baier & Katoen, 2008, p. 156) that if A = A A2, then L( A) = L( A1) ∩ L( A2). The Büchi automaton for the LTL formula ⫬Φ with Φ = G ⫬ β is shown in Fig. 5B; its synchronized product with the Büchi automaton in Fig. 5A is shown in Fig. 5C. The product BA has always a single initial state (spre, q0). The set of accepting subsets F for the example is F={F1}, with F1 = {(s, q1), ∀s ∈ RS}. Since, indeed, there is an infinite path that starts in (spre, q0) and that visits infinitely often at least one state in F1, then L ( M⊗A) is not empty and therefore the initial state of the net does not satisfy the LTL formula G ⫬ β. Note that the initial state of the net is the successor of the initial state of M⊗A.

From LTL to Sat∃LTL

A limitation of Algorithm 1 is that it only computes if the initial state minit satisfies the formula Aφ. For CTL ∗, we shall need instead the entire sat-set. Moreover, we switch to existentially-quantified LTL formulae, since they are easy to compute in symbolic form, noting that a state (s |= Aφ) ≡ (s |=⫬ E ⫬φ). In this section we describe the extension of Algorithm 1 for sat-set computation as Sat∃LTL(M, φ).

A remarkable result of Clarke, Grumberg & Hamaguchi (1997), Theorem 2, p. 54) shows that LTL model checking can be actually performed using a fair CTL model checker, not on the Kripke structure M but on the product GBA M⊗A. In particular, only a single CTL operator, Efair G true, is needed to check arbitrary LTL formulae.

We briefly recall the definitions of CTL fairness. A fairness condition F is a subset of states that must be visited infinitely often. A path π∈PM∞(s) satisfies a fairness condition F if the states of F appear infinitely often on π. A fairness set F is a set of fairness condition F, that must all be met together. A path π is a fair path w.r.t. F iff π statisfies all fairness conditions of F. A state s satisfies the fair CTL formula Eϕ w.r.t. fairness set F iff (1) it satisfies the CTL formula Eϕ, and (2) ∃π∈PM∞(s) that is a fair path w.r.t. F.

A state s satisfies a quantified LTL formula Eφ if in the product GBA M⊗A there is a path from s, q0 that visits infinitely often the states in the acceptance sets F, i.e. the same definition of the fairness condition acceptance in fair CTL. Thus if we consider the GBA as a Kripke structure, finding the states that satisfy Eφ becomes equivalent to finding the locations that originate fair paths, i.e. finding the states on the equivalent Kripke structure that satisfy Efair G true w.r.t. the fairness set F. The definition of Sat∃LTL(M, φ) that performs LTL model checking using fair CTL is described in Algorithm 2.

Algorithm 2 Sat-set of a quantified LTL formula Eφ.

1: procedure Sat∃LTL(M, φ)	
2:   L(M⊗A) ← GBA of φ	
3:   A ← BA of Kripke model M(RG), with S0 = RS	
4:   M ← synchronized product of A with M.	
5:   A is the set of the acceptance sets of F	
6:   Z0←succ(Q0M⊗A,δM⊗A)	
7:  M( M) is the Kripke model of A	
8:  switch type( A) do	
9:   case weak:	
10:    AS ← SATCTL(M( A), EF EG F1)	
11:   case terminal:	
12:    AS ← SATCTL(M( A), EF F1)	
13:   case otherwise	
14:    AS ← SATEFAIRG(M( A), true, fair = A):	
15:  return map (AS ∩ Z0) over RS	
16: end procedure	

The procedure first translates the path formula φ into a GBA A (line 2), it then builds the BA M of the input model M (line 3), described by its RG. Note that, since the goal is the sat-set computation, we need to consider all states as possible initial states, by taking S0 = RS in definition 7. Figures 6A and 6B show the BA M built from the RG of the net of the previous example, and the BA A for the formula Fβ. The algorithm then builds (line 4) the GBA M⊗A, as in Definition 8. The M⊗A automaton is then translated into the Kripke model M( M⊗A) (line 6 and 7). This translation is trivial due to the particular structure of M⊗A, since all arcs that enter a state carry the same set of atomic propositions, that can then be associated to the state itself. The sat-set of the CTL formula Efair G true for M( M⊗A) is then computed (lines 8 to 14), which is then mapped back to the states of the initial model (the RS of the Petri net) in line 15.

Figure 6 The Büchi automata built by Algorithm 2 for the sat-set computation of ELTL formulae.

The M⊗A of the example is shown in Fig. 6C: there is a single initial state (spre, q0) and the accepting set is F={F1}, where F1 = {(s,q1), ∀s ∈ RS}, therefore, in this case, M⊗A is a BA. The Kripke model M( M⊗A) of the synchronized product M⊗A is shown in Fig. 6D, together with all the sets computed by Algorithm 2.

Some aspects of this procedure need a deeper explanation, in particular the three different ways of computing the sat-set (lines 8–14) and the Z0 construction and its use (lines 6 and 15).

[Lines 8–14] This part computes the set of states that originate an infinite path accepted by the M⊗A BA. Line 14 is the general case: the results in Burch et al. (1992) and Clarke, Grumberg & Hamaguchi (1997) show how to compute the sat-set through the model-checking of the fair CTL formula EfairG(true,fair=F) on M⊗A. The procedure for the sat-set computation of EfairG(true,fair=F) for a Kripke model is reported in Algorithm 3. Line 10 and 12 correspond to two specific cases, as it was shown (Bloem, Ravi & Somenzi, 1999) that if M⊗A is a weak or terminal BA, the computation of Efair G true can be simplified.

Algorithm 3 Emerson-Lei algorithm for EfairGψ on Kripke model M and fairness constraints F.

1: procedure SATEFAIRG(M, ψ, F)	
2: S ← Sat(ψ)	
3: repeat	
4:   S′ ← S	
5:   for each Fi ∈ F: do	
6:    Y ← SATCTL(M, E (S U (Fi ∩ S)))	
7:    S ← S ∩ SATCTL(M, EX Y))	
8:   end for	
9:  until S = S’ // Repeat until fixed point is reached.	
10: return S	
11: end procedure	

A BA is weak if (1) it is possible to find a partition {Qi} of its set of locations Q so that each Qi is either contained in the accepting set or it is disjoint from it and (2) the {Qi} are partially ordered s.t. the transitions of the automaton never move from Qi to Qj unless Qi ≤ Qj. In this case the only way for a path to visit infinitely often a state of the accepting set F1 is to eventually be confined inside one Qi⊆F1, and an accepting run is a witness for the CTL formula EF EG F1. A terminal BA is a weak BA in which the {Qi} are maximal elements in the partial order. In this case an accepting run is a witness for the CTL formula EF F1. The time complexity for model checking the two CTL formulae is linear in the size of the model, while the cost for the general case in line 14 is instead quadratic. Note that Algorithm 2 tests the weak and terminal condition on A, and not on the (usually) much larger M⊗A, as it was proven in Bloem, Ravi & Somenzi (1999) that if A is weak (terminal) so is M⊗A. When this is the case the simplified procedure obviously applies also to M( M⊗A), as we do.

[Lines 6 and 15]. Z0 computed in line 6 is the set of immediate successor of the initial state(s) of the M⊗A. This step is needed because of the fictitious spre state introduced in M by the translation of Definition 7. The map operation in line 15 maps pairs (si, qj) onto the marking of si.

The sat-set of the EFβ for the Petri net system of the example is also listed in Fig. 6D. Note that (s3, q1) |= EfairG true, as there is indeed an infinite loop on the (s3, q1) state, which is an accepting state. But s3 corresponds to the Petri net marking m3, a deadlock state that does not satisfies β, so certainly s3⧸⊨EFβ. Indeed (s3, q1) ∉ Z0 since the witness path of EfairG true for (s3, q1) corresponds to a path in the automata of the formula that does not start from the initial location q0.

The work in Emerson & Lei (1987) gives a symbolic algorithm (known as Emerson-Lei algorithm) for model checking fair CTL properties, while (Burch et al., 1992) provides its fixed point characterization. Its time complexity is quadratic in the size of the automaton being checked. Algorithm 3 is an high-level view of the this fixed point characterization for the sat-set computation of EfairGψ formulae for a Kripke model M, with a set of fairness constraints F. In the algorithm, the call to SATCTL(M, Φ) returns the sat-set of the CTL formula Φ (without fairness constraints) on the Kripke model M. The Until and next path operators at line 6 and 7 use sets instead of atomic propositions or state formulae, for simplicity.

CTL ∗ model-checking procedure

To understand the CTL ∗ model checking procedure of starMC, we have to explain how a single CTL ∗ formula is divided into multiple quantified LTL formulae that can be checked using Sat∃LTL only. It may be worth starting from an example. Let’s consider the CTL ∗ formula

Ψ=EFGE(αU(AXβ))

where α,β are atomic propositions, and define the following (sub-)formulae:

Ψ1=AXβΨ2=E(αU(AXβ))Ψ3=E(αUaΨ1)Ψ4=EFGaΨ3

Let Sat(Ψ) be the sat-set of Ψ. Assume that we have a model checking procedure SATLTL(Ψ) that computes Sat(Ψ) if Ψ is a quantified LTL formula. Similarly, assume that a procedure SATCTL(Ψ) computes Sat(Ψ) if Ψ is a CTL formula. Formulae Ψ1 and Ψ3 can be considered as both quantified LTL and CTL formulae, while Ψ2 is not LTL, and Ψ4 is not CTL. If we assume that aΨi is an atomic proposition that holds in state s iff s |= Ψi, then we can compute Sat(Ψ) using only SATLTL on the sequence of formulae: Ψ1, Ψ3 and Ψ4. Vice-versa, Sat(Ψ) cannot be computed using only SATCTL, since SATCTL(Ψ2) allows to rewrite Ψ into Ψ4 which is an LTL formula for which it is well-known that no equivalent CTL one exists. According to Emerson & Lei (1987), the idea of substitution was already present in Emerson & Sistla (1984) and it was later used in Visser & Barringer (2000).

Algorithm 4 is the logical view of the CTL ∗ model-checking procedure implemented in starMC. The algorithm follows the ideas in Emerson & Lei (1987), and builds a CTL ∗ model-checker based on SAT∃LTL. It contains one case per type of state formula (by definition a CTL ∗ formula is a state formula). The only non-trivial case of Algorithm 4 is Eφ in line 7: if φ is an LTL path formula, SAT∃LTL can be directly applied to the formula, if instead φ contains additional quantifications, we first have to rewrite φ as an LTL path formula (by substituting quantified sub-formulae with newly created atomic propositions). Both cases are treated in a unified manner through the call of SAT∃LTL(REWRITE(φ)). Each formula identified for the substitution is a Maximal Proper Quantified Sub-formula (MPQS). A MPQS is defined in terms of the so-called Maximal Proper State Sub-formula (MPSS); see (Baier & Katoen, 2008), def.6.86.

Algorithm 4 CTL ∗ model-checking algorithm.

1: procedure SatCTL ∗(M, Ψ)	
2: switch Ψ do	
3:   case a ∈ AP: return {s ∈ SM | s |= a}	
4:   case Ψ1 ∧ Ψ2:	
5:     return SatCTL ∗(Ψ1) ∩ SatCTL ∗(Ψ2)	
6:   case ⫬Ψ: return SM \ SatCTL ∗(Ψ)	
7:   case Eϕ: return Sat∃LTL(M, rewrite(ϕ))	
8: end procedure	

Definition 9 (Maximal Proper State Sub-formula). State formula Ψ is a MPSS ofρ whenever Ψ is a sub-formula of ρ that differs from ρ and that is not contained in any other proper state sub-formula of ρ.

Let MPSS(ρ) be the set of all maximal proper state sub-formula of ρ.

Example (from (Baier & Katoen, 2008)). If ρ=Eφ and

φ=X(AGEFa)∧FGE(Xa∧Gb)

then MPSS( φ)={AGEFa,E(Xa∧Gb)}. Back to the initial example, MPSS(Ψ)={Ψ2} and MPSS(Ψ2)={Ψ1}.

Let’s consider this second example:

Ψ=EFGE(αU(α∧AXβ⏟Ψ3)⏟Ψ2)⏟Ψ1

than Ψ2 is a MPSS of Ψ1 and Ψ3 is a MPSS of Ψ2. In our model-checking procedure we consider instead a refinement of the MPSS definition, that we call Maximal Proper Quantified Sub-formula (MPQS). The motivations for introducing MPQS with respect to previous work and MPSS is given in “MPSS vs. MPQS”.

Definition 10 (MPQS). A Maximal Proper Quantified Sub-formula (MPQS) Ψ of a formula ρ is a MPSS of ρ which is a quantified state formula, i.e. either Ψ=Eϕ or Ψ=Aϕ.

The MPQS decomposition maximises the length of path formulae whenever possible. The MPQS decomposition of the example CTL ∗ formula is:

Ψ=EFGE(αU(α∧AXβ⏟Ψ3))⏟Ψ1

The state sub-formula Ψ2 cannot be a MPQS of Ψ1,since Ψ2 is not a quantified state formula.

REWRITE is the procedure to identify MPQS; its logical view is described in Algorithm 5, which is based on the syntactical recognition of the input formula ρ. Operators and atomic propositions are left unchanged (lines 3–9), when ρ=Eϕ (line 10) the formula ρ is rewritten as the atomic proposition aρ and to appropriately assign aρ, the sat-set of ρ is computed through the indirect recursive call to SatCTL ∗(ρ).

Algorithm 5 Identification and replacement of MPQSs.

1: procedure REWRITE(ρ)	
2:   switch ρ do	
3:      case ρ1 ∧ ρ2:	
4:         return REWRITE(ρ1) “∧” REWRITE(ρ2)	
5:      case ⫬ρ: return “⫬” REWRITE(ρ)	
6:      case Xρ: return “X” REWRITE(ρ)	
7:      case ρ1 U ρ2:	
8:          return REWRITE(ρ1) “U” REWRITE(ρ2)	
9:      case a ∈ AP: return “a”	
10:      case Eϕ:	
11:       // ρ = Eϕ is a MPQS. Replace ρ with an AP aρ.	
12:       AP ← AP ∪{new aρ}	
13:       Sat ← SATCTL*(ρ)	
14:       for each s ∈ Sat do	
15:          L(s) ← L(s) ∪ {aρ}	
16:       end for	
17:       return “aρ”	
18: end procedure	

Previous work

While the main sources of inspiration for the CTL ∗ model checking procedure presented in “StarMC: The Logical View” have already been introduced, in this section we discuss and compare with previous work with a broader view. We shall first discuss the different approaches to CTL ∗ model-checking and report on past and existing CTL ∗ model-checkers. Since the CTL ∗ model-checking procedure of starMC is based on LTL model-checking, we also provide a summary of the main approaches to LTL model-checking.

CTL ∗ model-checking approaches

CTL ∗ model checking procedures can be distinguished depending on whether they are conducted on the model itself or on an extended model (as in case of the synchronized product).

Model-checking the model itself

An example of this approach is when the CTL ∗ formula is first translated into μ-calculus and then the μ-calculus model-checking procedure is applied. This can be just a straightforward implementation of the fixed-point definition of the full μ-calculus or the more efficient algorithm proposed in Emerson & Lei (1986) for μ2-calculus, based on the observation that the formula translated from CTL ∗ belong to the μ2 fragment of the μ-calculus. The efficacy of symbolic techniques for model checking of μ-calculus was shown in Burch et al. (1992). In this approach the whole CTL* formula is translated into a single μ-calculus property: this translation is exponential in the size of the formula (Bhat & Cleaveland, 1996). A linear translation from of CTL ∗ to first order μ-calculus, a form of μ-calculus with data, is proposed in Cranen, Groote & Reniers (2011). Note that, in this case, a CTL ∗ formula may not produce a single first-order μ-calculus expression.

Model-checking an extended model

The model is extended to incorporate the formula information. The extended model is therefore constructed ad-hoc for each formula being checked. This approach may be further refined according to how the formula structure is encoded/exploited. Recursive descendent approaches based on formula substitution. As in the original papers by Emerson & Lei (1985, p. 90) and Emerson & Lei (1987), sub-formulae are identified and substituted, thus reducing the sat-set computation for CTL ∗ to a sequence of sat-set computations for LTL. The complexity of this CTL ∗ model-checking procedure is of the same order as that of LTL. It is well-known (Baier & Katoen, 2008) that for CTL ∗, as for LTL, the model-checking problem is PSPACE-complete, with a cost linear in the size of the system and exponential in the size of the formula (because of the Tableau or the Büchi automaton construction).

Hesitating alternating tree automata (HAA). A different approach is presented in Visser & Barringer (2000), where CTL ∗ formulae are translated into HAA. It is mentioned that to build the HAA for the CTL ∗ formula, when an ∃φ sub-formula is encountered it is necessary to build the Büchi automata that accepts all infinite words recognized by φ. The synchronized product of the HAA of the formula with the Kripke model is again an HAA, and checking that the original formula is satisfied on the Kripke model corresponds to checking that the language of the synchronized product HAA is empty. Also this approach is based on the idea of MPSS (or MPQS) substitution with a label. Although described in a different manner, it seems that the approach based on HAA shares many similarities with the approach described in “StarMC: The Logical View”. It basically does in an implicit manner what the SatCTL ∗ in Algorithm 4 does in an explicit one. In the same paper the authors also propose a model checking procedure based on games, where the objective is to find a winning strategy for the non-emptiness game on the synchronized product of the Kripke model and the HAA of the formula (Theorem 2 in Visser & Barringer (2000)).

Sat-set computation for EfairG Ψ

The work in Emerson & Lei (1986), already mentioned for its contribution in proving that μ2-calculus can be efficiently model-checked, also provides a succinct translation from fair-CTL to μ2-calculus, paving the way to efficient model checking of fair-CTL, in particular for the sat-set computation of Efair GΨ. The algorithm is provided in an abstract form, regardless of the specific data structures choice. It is readily suitable for a symbolic implementation, which is what we have used for starMC.

A different symbolic solution for the computation of a fair cycle is provided by the OWCTY algorithm (Černá & Pelánek, 2003), which does not compute sat-sets but aims at building counter-examples as quickly as possible. A comparison of the two approaches for LTL model checking can be found in Duret-Lutz et al. (2011).

Sat-set computation for ELTL

Buchi-based model checking of LTL of a formula Φ is defined in Vardi & Wolper (1986), as recalled in “StarMC: The Logical View”. The paper in Burch et al. (1992) shows a symbolic model checking procedure for LTL. It is based on an implicit tableau construction and on its symbolic representation. It is then shown that the sat-set of an LTL formula can be reduced to the fair CTL symbolic model-checking for the μ-calculus expansion of EG true, under appropriate fairness constraints.

A later work (Clarke, Grumberg & Hamaguchi, 1997) encodes the tableau of the LTL formula along with the model variables in a single SMV (Clarke et al., 1996) model, and explicitly reduces the LTL model checking to a fair-CTL formula, using the existing and unmodified CTL model checker of SMV. Model checking fair CTL is based on the Emerson-Lei algorithm (Emerson & Lei, 1986). This procedure, together with its implementation in BDD is well explained in the survey paper by Rozier (2011). The survey also cover many other related topics, including a comparison of the expressiveness of LTL, CTL and CTL ∗ both in theory and in practical applications.

starMC exploits the Buchi-based approach in Vardi & Wolper (1986), that makes the synchronized product of the Büchi automaton and the model, and uses the algorithm in Emerson & Lei (1986) for the detection of the accepting runs and the consequent construction of the set of states that satisfy the formula. The library Spot (Duret-Lutz, 2014; Duret-Lutz et al., 2016) provides different, efficient, ways to build a Büchi automaton from an LTL formula. The generated automaton may have state-based or transition-based acceptance. This latter choice, based on the translation defined in Couvreur (1999), implemented in Spot using decision diagrams, is usually the most efficient. starMC nevertheless uses state-based automata, since it allows an easier re-use of the SatCTL module of GreatSPN.

A number of possible optimizations have been presented in the literature for LTL model-checking. For example in Duret-Lutz et al. (2011) a modified algorithm called Self-Loop Aggregation Product (SLAP) is proposed, where instead of building a synchronized product in an extended space (model and automaton), the joint state is kept into separate aggregates. Accepting runs are then searched only in those aggregates that may contain cyclic runs over the accepting states. For the time being starMC does not include this kind of algorithmic optimization, although a certain attention has been taken to avoid inefficiencies in the symbolic implementation.

We do not review here the numerous techniques for the on-the-fly model-checking of LTL, since these techniques concentrate on establishing the truth value of a formula in the initial state, while the Sat∃LTL procedure needed for CTL ∗ model checking requires the construction of the sat-sets of the LTL sub-formulae.

CTL ∗ model checkers

According to a relatively old paper (Barringer et al., 2002) there was a CTL ∗ model-checker available in the Rainbow verification system (Barringer et al., 2002): the tool is not available any longer and we could find no published details on how the model checker was built, only that it was based on HAA. Previous papers by the same authors evaluated whether it was the case to enhance the tool SPIN with CTL ∗ (Visser & Barringer, 1998; Visser & Barringer, 2000), but apparently it was never done and/or reported in the literature.

There is an implementation of μ-calculus for the nuXmv tool (Cavada et al., 2014), which, according to its website, could lead to a CTL ∗ model-checker but, for the time being, only CTL ∗ formulae that are either LTL or CTL are actually processed.

The LTSmin tool (Kant et al., 2015) includes a CTL ∗ model checker. It allows to model check a Petri net by translating the CTL ∗ formula to be checked into an equivalent μ-calculus one, using the procedures defined in Dam (1990). A model checker of μ-calculus for Petri nets is available also in TINA (Berthomieu, Ribet & Vernadat, 2004), but no translator from CTL ∗ to μ-calculus is provided.

MPSS vs. MPQS

In the literature there have been slightly different definitions of the CTL ∗ sub-formulae to be substituted and, although they are all correct and adequate for CTL ∗ model-checking based on Sat∃LTL, different definitions may lead to a different number and/or to a different structure of the Büchi automata being built. The work in Emerson & Lei (1987) does not contain a definition in strict sense of the formulae to be substituted, but it states that, for each formula Eϕ, where ϕ is a path formula, then the set of formulae to be substituted by atomic propositions are the “top-level proper existential sub-formulae Eqi of ϕ that are not sub-formulae of any other sub-formula Er of Eϕ, where Er is different from Eϕ and from Eqi”. The Eqi are therefore the largest quantified sub-formulae of the formula ϕ.

In later times the substitution of formulae with atomic propositions was based on the definition of MPSS (Visser & Barringer, 2000), which is equivalent to the one given by def.6.86 in Baier & Katoen (2008), reported in this paper as Definition 9. The definition of MPSS encounters two problems when deriving an algorithm, it may re-label more formulae than strictly necessary, and it may result in a non-deterministic algorithm since the CTL ∗ grammar is ambiguous. Indeed in the considered example:

Ψ=EFGE(αU(α∧AXβ⏟Ψ3)⏟Ψ2)⏟Ψ1

Since ψ2 is both a state and a path formula due to the ambiguity of the CTL ∗ language, there are two evaluation strategies: Evaluate Ψ1 as E (α Uaψ2);

Evaluate Ψ1 as E (α U(α ∧ aψ3));

Both strategies are correct, but they are not equivalent. The former strategy is based on the identification of MPSS sub-formulae. While correct, it inhibits possible reductions and optimizations for Büchi Automata generation, since the label aψ2 clearly carries less information than the (α ∧ aψ3) formula. This example also shows that MPSS are different from “top-level proper existential sub-formulae” which are at the base of the CTL ∗ model checking algorithm in Emerson & Lei (1987) and that we have formally defined as MPQS in Definition 10. The MPQS decomposition of the example CTL ∗ formula is:

Ψ=EFGE(αU(α∧AXβ⏟Ψ3))⏟Ψ1

The state sub-formula ψ2 cannot be a MPQS of Ψ1, since Ψ2 is not a quantified state formula.

Starmc: the inner view

This section provides an insight on the implementation of starMC, which consists in a DD-based implementation on the algorithms of “StarMC: The Logical View”. The implementation relies on the Meddly and Spot libraries, and on the symbolic CTL model checker already available in GreatSPN.

In starMC DDs are used to encode all data structures, but for the BA of the (sub-)formulae, that are instead represented in explicit form. Petri net state spaces, Kripke models and Büchi automata are all expressed in DD forms using the M×D of the NSF and the MDD of the states (when needed). Most computations of the model checker work with DDs that encode “potential” states. Potential states are a superset of RS that arise because the NSF, which is built before the RS, may include the firing of transitions also from states that are not reachable for the given initial marking. Algorithms (like state space generation) that work forward, from the initial marking onward, never generate unreachable states, but this is not the case in the implementation of some path operators, that works backwards (e.g. the neXt). Working on potential states is a standard choice for DD-based model-checkers, and intersections with the RS are done only when strictly needed.

CTL ∗ model checking

Implementation of SatCTL ∗ and REWRITE

We do not provide a detailed implementation view of Algorithm 4 for the sat-set computation for CTL ∗, but we simply identify the data structures used. Sat∃LTL returns an MDD, RS in line 6 is an MDD and the set operations in line 5 and 6 are also implemented as MDD operations. The value returned by REWRITE (ϕ) is an LTL expression following Definition 4. An implementation choice worth noticing is whether SM in line 3 and 6 of the algorithm refers to the MDD of the actual RS or of the set of potential states ( PS). This part is common to the CTL model-checker of GreatSPN, for which the default choice is to manipulate potential states. Intersection with reachable states is done only for negation (line 6) (set difference from RS) and for the sat-set of true, for which RS is used. Atomic propositions (line 3) are also defined on the PS.

Implementation of M construction

The generation of the decision diagrams that encode the M⊗A automaton is summarized in Algorithm 6. The MDD (M×D) of M⊗A is defined to have n + 1 (2n + 2 resp.) levels: one level each for the n places of the net and one for the location of A. We indicate with ⟨m,q⟩ a joint state of M⊗A, where m is a model state and q is an automaton location.

Algorithm 7 Implementation view of SAT∃LTL.

1: procedure SAT∃LTL(M, φ)	
2:    A ← translate φ into a Büchi Automaton	
3:    ⟨Z0,NSF,SF⟩ ← BUILDSYNCHPRODUCT(M, A)	
4:   S ← Saturate(Z0, NSF)	
5:   K = (S, NSF ) is a Kripke model	
6:   AS ← MDD()	
7:   switch type( A) do	
8:      case weak:	
9:       AS ← SATCTL AS←SATCTL(K,EFEG(SF[1]))	
10:     case terminal:	
11:       AS ← SATCTL AS←SATCTL(K,EF(SF[1]))	
12:     case otherwise:	
13:       AS ← SATEFAIR AS←SATEFAIRG(K,true,fair=SF):	
14:   Sat(∃φ) ← RemLoc(AS ∩ Z0)	
15:   return Sat(∃φ)	
16: end procedure	

The algorithm builds the MDD and the M×D of M⊗A taking as input a graph description of the formula automaton A and the M×D for the NSF of M. There is no MDD for the initial states of M since all states are considered as initial ones. The algorithm implements at the same time the translation of M into a Büchi automaton (Definition 7) and the M⊗A construction (Definition 8). The algorithm directly builds the Kripke model, without passing through the M⊗A BA as described in “StarMC: The Logical View”. The DD representation of the M⊗A automaton of Algorithm 6 consists of three elements: M×D NSF for the δ function; an array SF of MDDs of the accepting sets; and the MDD for the set Z0 of states as defined by Algorithm 2.

Figures 7B–7G shows the DDs built by Algorithm 6 for the net of Fig. 1A and for the formula expressed by the Büchi automaton in Fig. 7A. These DDs represent the symbolic encoding of the information (models and sat-sets) of Fig. 6.

Figure 7 (A–G) A Büchi automaton of a CTL ∗ formula and the DDs generated by its evaluation for the net in Fig. 1A.

Construction of Z0 (lines 4–8)

If q0 is an initial state of A with an outgoing arc (q0→aq′) then Z0 should include all states ⟨s,q′⟩ for all states s that satisfy a (s ∈ Sat(a)). The function AddLoc(d, q) takes a MDD d and sets the location level to q. Figure 7B shows the MDD of Z0.

Construction of the NSF (lines 9–14)

The M×D for NSF is built by modifying the M×D NSFM of the input model M to include only “appropriate states” of M and to add the locations of A. For each edge q→aq′ of A a new M×D is created (line 12), by modifying the NSF of M to reach only SatM(a) markings (the states of M that satisfies a), and at the same time by moving the GBA location from q to q′. Function AddLocX(v, q, q′) takes a M×D v and adds a relation q→q′ for the first primed and unprimed levels (automata locations). The transition relation δ of M⊗A is encoded as a NSF from the union of all the edge M×Ds (line 13). Figure 7C shows the M×D of the NSF of the M⊗A of the example.

Identification of the array of accepting subsets SF (lines 15–19)

The set SF is simply implemented by the array SF of MDDs, one entry per Fi. According to Definition 8, each accepting subsets Fi is the Cartesian product of FiA the i-th subset of final locations of A with all the states of M. The Cartesian product Fi can be realized by adding one level on top of an MDD v that encodes the states of M. The node of this top level has a down-arrow to v for each location q∈FiA. A more efficient solution is actually implemented in Algorithm 6, based on the observation that we can consider “potential” states of M by considering any combinations of tokens in any place. The corresponding MDD, in fully reduced form, is very compact: it has n + 1 levels, the node of the top level carries the FiA elements, all other levels are skipped. This MDD can be built by taking (line 17) the union of MDDs created by the Meddly function edgeForVar(q, n + 1), which creates a fully reduced MDD in which level n + 1 is set to q and all other levels are skipped.

Algorithm 6 Synchronous product construction between M=⟨RSM,NSFM⟩ and A=⟨Q,AP,δ,Q0,F⟩.

1: procedure BUILDSYNCHPRODUCT(M, A)	
2:   // Build the MDD of the initial states.	
3:   Z0 ← MDD()	
4:   for each location q0 ∈ Q0: do	
5:     for each edge e=q0→aq′ in δ: do	
6:        Z0 ← Z0 ∪ AddLoc(SatM(a), q′)	
7:     end for	
8:   end for	
9:   // Build the Next State Function M×D	
10:  NSF ← M×D()	
11:  for each edge e=q→aq′ in δ: do	
12:     nsf e ← AddLocX(NSFM ∩ (PSM × SatM(a)), q, q′)	
13:    NSF ← NSF ∪ nsfe	
14:  end for	
15:  // SF: array of MDDs, one entry per accept. set Fi ∈ F	
16:  for each accepting set Fi ∈ F: do	
17:     MDDFi←∪{edgeForVar(q) | for each q∈Fi}	
18:     SF[i]←{MDDFi}	
19:  end for	
20:  return ⟨Z0,NSF,SF⟩	
21: end procedure	

In the automaton of the example F={F1}, with F1 = {q1}, therefore SF is an array with a single element, the MDD that combines the value 1 (encoding of q1) for the location level with any combination of tokens in all places. The MDD is depicted, in fully reduced form, in Fig. 7D.

Implementation of Sat∃LTL

Sat∃LTL computes the set of states of M that satisfy the quantified LTL formula Eφ. Algorithm 7 provides an implementation view of Algorithm 2. Line 2 is a call to Spot for formula φ, that builds a Büchi automaton in textual HOA form. The DDs of M⊗A are built by the call in line 3. The set of reachable states S in M⊗A is then generated in line 4 using saturation (Ciardo, Lüttgen & Siminiceanu, 2001).

Figure 7A is the automaton built by Spot for the formula Fβ, to be evaluated for the net in Fig. 2A, with β = (#P1 > 0 or #P2 > 0)}. Figures 7B–7D are the three DDs returned by the call of BuildSynchProduct in line 3, while Fig. 7E is the result of the Saturation procedure on M⊗A.

Note that in the M⊗A construction of Algorithm 6 all informations on edge labelling is lost, since it is now irrelevant, and, to find the set of accepting states (AS) of the formulae in lines 9, 11, and 13, we can consider ⟨S, NSF ⟩ as a Kripke model (line 5). Lines 7 to 13 follow the same logic of Algorithm 2.

As for Algorithm 2 the computed AS set is not the accepting set of the original formula: the operations in line 14 build the MDD intersection of the MDDs of AS and Z0 and applies to the resulting MDD the RemLoc function, that removes the DD level that stores the location index, thus obtaining an MDD that encodes the reachable states of the Petri net model that satisfy Eφ.

The MDD of the AS set for the example is given in Fig. 7F. It has been computed by the call to the CTL model-checker in line 11, since A is a terminal automaton. Note that DD represents AS in the potential state space. Moreover not all the accepting states in AS are in the sat-set of Eφ, either because they are not reachable (are in PR\RS) or because they are not “aligned” with the initial conditions of A, otherwise said, they have to belong to Z0. The result of the intersection and RemLoc operations in line 14 is shown in Fig. 7G, that encodes the 3 satisfying markings (all markings of RS except the one where all places have zero tokens). Note that M⊗A states (q0, 1, 1, 0) and (q1, 0, 0, 0), are in the DD of the AS but nor (1, 1, 0) nor (0, 0, 0) are in the DD of Sat( E Fβ). The first one is a state which is not in the RS of M⊗A, the second one is not in Z0. Both are removed by the intersection in line 14.

Implementation of EfairG

The logical view for the sat-set computation of EfairG given in Algorithm 3 is quite close to its DD-based implementation. The EfairG is called by Algorithm 7 on a Kripke model encoded with an M×D for its NSF, a MDD for its states, and an array of MDDs for the set of fairness constraints F. All DDs are for n + 1 variables, the n places of the Petri net model and the location of the automaton of the formula. The SatCTL calls in line 6 and 7 of Algorithm 3 have been implemented with the pre-existing DD-based CTL model checker of GreatSPN, modified so as to treat n + 1 levels. The intersections in lines 6 and 7 are straightforward MDD intersections.

Testing results

We have tested the correctness of the results and the performance of the tool. All tests were performed using the publicly available (mcc.lip6.fr/2019/archives/2019-mcc-models.tar.gz) MCC2019 benchmark (Kordon et al., 2019). The MCC2019 benchmark includes 1,018 model instances, that are parametric variations (in structure or in the initial marking) of 94 distinct colored and P/T models. There are both academic (e.g. Philosophers, Kanban, Erathostenes, etc.) as well as relevant industrial models (ARM processor cache, biochemical networks, UML models, CAN bus models, etc.). For each model instance different properties are defined: basic Petri net properties (like cardinality of the state space, absence of deadlock, boundedness and liveness), 32 LTL properties and 32 CTL properties, for a total of about 60 k LTL and CTL formulae. The benchmark includes all known formula evaluations (truth value in the initial marking), making MCC data a valuable benchmark for (Petri net) tools. The tests use Spot version 2.9.6, the latest Meddly (tag 7a31ca8 on gitHub), LTSmin version 3.0.2. and GreatSPN (tag 4439bde).

The semantics of LTL and CTL is defined over infinite paths, but Petri net models with deadlocks also feature finite paths. For MCC, CTL is directly defined on RG and not on Kripke structures, hence the successor function may not be total. In that case, the CTL semantics is axiomatically defined (Lichtenstein, Pnueli & Zuck, 1985) for the set Sd of deadlock states as: EX Sd = Ø, and EG Sd = Sd. LTL and CTL ∗ properties are interpreted considering a proper Kripke structure (i.e. stuttered) built on the RG according to Def. 2.

The benchmark does not include any CTL ∗ property for the MCC models (and for any other models we could find). Therefore we have algorithmically generated a set of CTL ∗ formulae from the MCC CTLCardinality queries by randomly deleting path quantifiers with a probability of 70%. The top-most quantifiers are always kept, in order to preserve consistency with the CTL ∗ grammar. The sat-set cardinalities of the LTL and CTL formulae are also missing in the benchmark. We have therefore conducted the comparison tests on the cardinalities of the produced sat-sets by checking consistency between multiple tools.

Resource setting. All experiments have been performed on dedicated cpu cores (Intel Xeon CPU E5-2630 v3 at 2.40 GHz) with a limit of 2 GB of memory and a 60 s timeout for state space generation and another 60 s to check each formula. Variable orders were pre-computed using the state-of-the-art heuristic of Amparore, Donatelli & Ciardo (2020a).

Examined models. Within the 60 s limit, starMC is able to build the RS of 434 model instances, from 77 different models. This set includes many very large model instancess, both in terms of places and transitions (up to thousands) and number of states. Figure 8 shows a chart of the distribution of the sizes of the 434 model instances. Two of these instances have a state space of the order of 10478, one in the order of 10603 (numbers not even representable in a double precision floating point). The 67 model instances with more than 1021 states (the two rightmost columns in Fig. 8) correspond to 29 different models.

Figure 8 Distribution of the state space size of the 434 model instances built by starMC in less than 60 s.

Analysis’ objectives. The characteristics that have been tested are: CT: Correct Truth value of the formula for the initial state

CC: Correct Cardinality of the sat-set of the formula

PE: PErformance (execution times)

CT test based on MCC results

starMC results are checked against the available truth values for CTL and LTL properties. For each instance we computed the result for 32 CTL and 32 LTL properties, for a total of 27,776 queries (434 ∗ 64) queries. With the test resource limitations, starMC was able to terminate about 80% of the queries. All unanswered queries did not finish in the time bound. None of the query evaluation generated a not-enough-memory error. For 7 of these 434 models, the query format was corrupted due to bad namings. These 7 models were thus dropped. We found no mismatch w.r.t. the MCC published values for the remaining 427 models.

To get a better accuracy we need to verify more precise information like the sat-set cardinalities (the CC test), as in the next steps, but before proceeding to sat-set computation we have investigated the possible overhead caused by the (exponential) size of the Büchi automata. We have computed the size of the Büchi automata for about 14 k LTL queries. The largest automata produced by Spot has 70 locations and 356 edges. The average number of locations (edges) is 3.95 (7.7). This correspond to a maximum and average number of path operators equal to 12 and 4.7 respectively. Note that starMC may build more than one Büchi automata per formula when model-checking CTL and CTL ∗ formulae, due to nesting.

CC and PE tests based on RGMeDD

This test compares the results of starMC and RGMeDD on CTL formulae. RGMeDD is the CTL model checker (Amparore, Beccuti & Donatelli, 2014) of GreatSPN, based on the standard recursive sat-set computation of CTL. The test therefore compares two different approaches to CTL model checking: a standard fixed point implementation (RGMeDD), and an automata-based implementation (starMC). Both tools use the same library for DD manipulation, moreover the same variable orders have been used. The benchmark is again on 427 instances, for a total of 13,664 CTL queries (427 ∗ 32).

Table 1 summarizes the behaviour of the two tools, while the plots in Fig. 9 report the execution times in linear form Fig. 9A and log form Fig. 9B: dots below the diagonal are queries for which RGMeDD is faster than starMC and vice-versa. Executions that have timed out are marked as TO.

Figure 9 (A–B) Execution times of starMC vs. RGMeDD, on 13,664 CTL queries.

Table 1 Summary of the experiments with starMC and RGMeDD, on 13,664 CTL queries.

Characteristics	Value	Characteristics	Value	
No. of queries	13,664	Same Time both	317	
starMC terminates	10,223	Both timed-out	2,447	
RGMeDD terminates	10,740	Only RGMeDD timed-out	476	
Both terminate	9,747	Only starMC timed-out	993	
starMC faster	938	starMC timeout and RGMeDD out of memory	1	
RGMeDD faster	8,492	Mismatches in sat-set cardinality	0	

RGMeDD completes 10,740 queries (79%) from 427 different model instances, while starMC completes 10,223 queries (75%) from 399 different model instances. All the 9,747 queries completed by both tools (71%) produced sat-sets of equal cardinality (CC test).

We have also analyzed the different behaviour in terms of solved queries and model instances, with RGMeDD being able to solve more queries but especially for more diverse instances.

Moreover, on the queries completed by both tools (blue dots in the plot), RGMeDD was faster than starMC on 87% of the queries, slower on the 10% and equal time (up to millisec) on 3%. There are a number of cases in which the execution times are rather different (values close to only the x or the y axis). This result for the PE test is somehow unexpected: considering that the same DD library is used, and the same variable orders, the differences are possibly due to the different model-checking approaches, and we actually expected the standard CTL procedure of RGMeDD to be more efficient in almost all practical cases. Factors that could make starMC faster than RGMeDD are the optimization of the formula done by Spot when building the Büchi automaton and a different use of the potential state space in the two implementations.

Tests based on LTSmin

The last tool assessment is based on a comparison with LTSmin (Kant et al., 2015): CC and PE are assessed for LTL, CTL and CTL ∗ formulae. LTSmin is run using the pins2lts-sym interface with the –ctlstar option, which first converts the input formula into μ-calculus and then applies the μ-calculus model-checker of LTSmin. It is worth noting that this translation may incur an exponential cost since LTSmin uses the translation described in Dam (1990), which is in theory less efficient than using Büchi automata. LTSmin is also based on decision diagrams, provided by the multi-core library Sylvan (van Dijk & van de Pol, 2017). To make a meaningful comparison, for each model instance we have enforced the use of the same variable order for the two tools. We have initially checked that both tools, when given the same variable order, produce exactly the same DD in output. Thus the choice of the variable order does not penalize one tool or the other, as long as it is the same for both tools. We give LTSmin the same time constraints as the other tools, i.e. 60 s to generate the state space, and 60 s to translate the formula in μ-calculus and perform the model checking. Since we have experienced inconsistencies of the LTSmin tool when dealing with models with deadlocks, we have considered deadlock-free models only. Moreover, only P/T models have been included in the analysis, since LTSmin does not treat colored models, leaving 222 model instances from 29 distinct models. For what concerns the type of queries, only LTL/CTL/CTL ∗ queries of the “Cardinality” category have been checked, since LTSmin cannot express atomic propositions based on the enabling of transitions. We did not consider model instances for which none of the two tools was able to compute the state space, leaving 123 model instances from 28 different models. Of the 123 models, LTSmin builds the state space of 86 and starMC of 116. In all tests, LTSmin ran on four cores (to allow it to exploit the parallelism of the Sylvan library) and starMC on a single one, similar to the settings used for the MCC competition.

The first test (CC and PE for LTL and CTL formulae) compares the sat-set cardinalities and the execution times for LTL and CTL formulae, while the second test (CC and PE for CTL ∗ formulae) extends the analysis to CTL ∗ formulae, the final target of the whole testing procedure.

CC and PE for LTL and CTL formulae

We computed the sat-sets generated by starMC and LTSmin (running in –ctlstar mode) on LTL and CTL formulae on the 123 model instances selected as explained above, with 16 CTL formulae and 16 LTL ones for each model instance. The objective is to check the performance of the two model checkers (PE) and, for all queries solved by both tools, check that the same sat-sets’ cardinalities are computed (CC). The results are summarized in Table 2, while a comparison of the execution times in log and linear form (as for the previous experiment) is reported in Fig. 10.

Figure 10 (A–B) Execution times of starMC vs. LTSmin on CTL and LTL queries.

Table 2 Summary of the experiments with starMC and LTSmin on CTL and LTL queries.

Characteristics	Value	Characteristics	Value	
No. of queries	3,936 (123*32)	Same Time both	1	
starMC terminates	2,990	Both timed-out	895	
LTSmin terminates	1,758	Only LTSmin timed-out	1,277	
Both terminates	1,708	Only starMC timed-out	50	
starMC faster	485	Mismatches in sat-set cardinality	7	
LTSmin faster	1,222			

PE results: in the 60 s limits, starMC solves 50% more queries than LTSmin, although, in the subset of queries solved by both, LTSmin is faster than starMC on 2 out of 3 queries.

CC results: of the 1,708 queries computed by both tools, we have a mismatch in the cardinalities of the sat-sets of 7 formulae (from different model instances). These are all LTL formulae. All results computed by starMC are consistent with the MCC known truth values. Since truth values of MCC are assigned according to majority of the output of the tools that participate in the competition, they may not be 100% reliable. We have therefore computed the sat-sets of the sub-formulae, which indicate that LTSmin is computing the wrong sat-sets, most likely due to a wrong translation to μ-calculus. For example, the sat-set of the formula named LTLCardinality-08 for the model BART-PT-002 has a structure “ X F G Fa”, where a is an atomic proposition and the formula is implicitly quantified for all paths. LTSmin reports that the formula has an empty sat-set, although the sat-set of the formula “ F G Fa” is equal to the full state space, which clearly indicates contradictory results. The translation in μ-calculus of the former query leads to a recursive formula with 7 μ and 18 ν fixed point operators, and it is not trivial to assess whether there is an error in the translation of the formula into μ-calculus or in the model-checking procedure. This behaviour has been reported to the LTSmin developers.

CC and PE for CTL ∗ formulae

The tests of “CC and PE for LTL and CTL formulae” are here repeated for the set of generated CTL ∗ formulae. The results are summarized in Table 3, while a comparison of the execution times in log and linear form (as for the previous experiments) is reported in Fig. 11.

Figure 11 (A–B) Execution times of starMC vs. LTSmin on CTL* queries.

Table 3 Summary of the experiments with starMC and LTSmin on CTL* queries.

Characteristics	Value	Characteristics	Value	
No. of queries	1,968 (123*16)	LTSmin faster	536	
starMC terminates	1,442	Both timed-out	505	
LTSmin terminates	775	Only LTSmin timed-out	688	
Both terminates	754	Only starMC timed-out	21	
starMC faster	218	Mismatches in sat-set cardinality	10	

PE: in the 60 s limits, starMC solves the highest number of queries (73%), almost twice those solved by LTSmin. In the subset of queries solved by both, LTSmin is faster than starMC on almost 4 out of 5 queries.

CC: of the 753 queries computed by both tools, we have a mismatch in the cardinalities of the sat-sets of 10 formulae (from different model instances). We have checked these formulae one by one, by computing the sat-set of sub-formulae, when meaningful and useful. In all but one case, the two tools differ radically: the sat-set being the whole state space for one and the empty set for the other, or vice-versa. In these cases it was not difficult to check that LTSmin is computing results that are inconsistent with the sat-sets of the sub-formulae computed by the tool itself. For the remaining case a special procedure was put in place that indicates that it is likely that LTSmin is computing the wrong result2 .

Conclusion

starMC is a CTL ∗ model-checker that computes the set of reachable states that satify a CTL ∗ formula. To the best of our knowledge it is the only available CTL ∗ model-checker based on Büchi automata for Petri nets, and also the only available CTL ∗ model-checker that does not require a translation into μ-calculus.

starMC provides CTL ∗ and LTL model-checking capabilities to the GreatSPN tool, improving the existing CTL model-checker. starMC is fully integrated into the GreatSPN GUI.

The implementation leverages two libraries: Spot for the translation from LTL sub-formulae to Büchi automata, and Meddly for decision diagram manipulation. The tool also includes the existing CTL model-checker of GreatSPN, modified to verify M⊗A structures.

Although many of the ideas behind the construction of starMC have been around for decades, there was not, as far as we know, a fully developed description on the algorithms for a (symbolic) implementation of a CTL ∗ model-checker, that we consider to be a significant contribution of this paper. The availability of such a tool is also very important: for educational purposes both in university and in industries, and for application in real-life contexts. The testing section has shown that, even on reduced resources (2 GB of memory and execution time with a 60 s time-out) starMC can model-check very large state spaces (we reached 10480 states) for formulae with equivalent Büchi automata of more than 70 locations and 350 edges.

Petri nets only?

starMC has been developed for Petri nets, either in GreatSPN format or in the PNML standard format. Although having a model-checker fully integrated into a specification and verification tool is an advantage for the user, it may hamper its reuse in the research community. There are indeed tools like LTSmin that try to be as general as possible, by providing a (formalism agnostic) intermediate language. starMC exploits Petri nets to collect useful information: computation of place bounds (so the user does not have to identify them beforehand) and heuristics for variable orders, which are based on P-semiflows (a structural property of the Petri net). Both issues relates to performance and do not hamper the application of the results presented in this paper to other formalisms for DEDS; nevertheless the tool is not currently structured for direct reuse as a stand-alone multi formalism model-checker.

Tool’s limitations

starMC builds the whole state space beforehand even for properties that could be proved by inspection of a part of the state space (no on-the-fly verification), and it does not build counter-examples.

Answer to research questions

(R1): our testing campaign has shown that it is possible to realize an efficient and fully symbolic implementation of the computation of sat-sets of LTL and CTL ∗ properties based on Büchi automata. (R2): the experiments conducted in the testing phase indicate that starMC can solve (significantly) more formulae than LTSmin, although, on the formulae solved by both tools, LTSmin is generally faster. (R3): an extensive comparison of different variable order heuristics has not been conducted yet, but the size of the system that we were able to solve show that, indeed, the variable order heuristic used, that was the best one for state-space exploration, as reported in Amparore et al. (2019), is at least adequate for CTL ∗ model-checking. (R4): the construction of counter-examples and witnesses has not been addressed in this paper.

Current and future work

starMC builds the whole state space beforehand even for properties that could be proved by inspection of a part of the state space (no on-the-fly verification). We are working on a model-checker that keeps the NSF in implicit form: the M×D of the NSF is substituted by a function that performs the firing of the Petri net transitions by directly manipulating the MDD of the state space being built. With a NSF in implicit form it is possible to implement an on-the-fly approach, in which the state space is built incrementally up to the point where the property can be (dis-)proven. The results could be extended to CTL ∗ following the work in Bhat, Cleaveland & Grumberg (1995), that develops an on-the-fly procedure for LTL and then extends it to CTL ∗.

No model-checker is fully useful if it does not produce meaningful counter-examples. GreatSPN already produces CTL counter-examples or witnesses for the initial state. Extending the current approach to CTL ∗ may require a definite shift in the approach, for example by looking into the creation of evidence for μ-calculus (Cranen, Luttik & Willemse, 2015), while keeping an eye on the efficiency of the construction, as in Jiang & Ciardo (2018).

Although CTL ∗ allows to express fairness constraints in the formula, we plan to develop also a model checker for the fair variant of CTL. Although this may seem a straightforward task, considering the already existing implementation of EfairG ψ, further analysis is needed to understand the most adequate form of fairness, in particular if we want to consider not only fairness based on visited states (and therefore among enabled actions), but also fairness of taken actions, to include, for example, the “Fairness running” clause of the NuSMV tool (Cavada et al., 2014).

The example in “CTL ∗ Model-checking Procedure” points out that a CTL ∗ model-checker could be realized based on a mixture of Sat∃LTL and SatCTL. Beyond the theoretical differences in complexity, the construction of such a CTL ∗ model-checker requires more experiments to understand if there is a practical advantage in doing so. Our comparison in “CC and PE Tests Based on RGMeDD” points out that it could be so.

Availability

A virtual machine, with the reproducible benchmark (scripts, models, tools, instructions) is available as a Zenodo permalink. The provided benchmark can be run either in quick or in full mode, which will take about 30 minutes/10 days, respectively, to complete. Instructions are provided inside the VM, available at: https://zenodo.org/record/5752419 with the name starMC-benchmark.ova.

A second virtual machine with the starMC tool pre-installed can also be found at the same link, with the name starMC.ova.

The source code of GreatSPN is available at https://github.com/greatspn/SOURCES.

The benchmark data is available at https://github.com/amparore/starMC-benchmark.

We would like to thank Jaco van de Pol for the various insights given on LTSmin, the researchers and developers of the Spot and Meddly libraries, and all researchers across the world that have contributed to the MCC benchmark.

Additional Information and Declarations

Competing Interests

Author Contributions

Data Availability

1 For simplicity we have used a P/T model, so the model is not parametric in the number of processes. To draw a parametric model, colored Petri nets can be used from the same interface, and a P/T model can be obtained by automatic unfolding.

2 There is a query of a model (named FMS-PT-00020 in MCC) for which the two sat-sets differ for a “small” portion: for LTSmin the sat-set is the whole RS, for starMC is a strict subset of RS (with a difference of 104 states out of the 1012 states in RS). Due to the complexity of model and formula we had to build a reduced model (applying standard Petri net reduction rules) with only 32 states, and a reduced formula, for which the two sat-sets differ by four states. In this case it was possible to check by hand that the 4 should not be in the sat-set, as computed by starMC.

The authors declare that they have no competing interests.

Elvio Gilberto Amparore conceived and designed the experiments, performed the experiments, analyzed the data, prepared figures and/or tables, authored or reviewed drafts of the paper, and approved the final draft.

Susanna Donatelli conceived and designed the experiments, analyzed the data, authored or reviewed drafts of the paper, and approved the final draft.

Francesco Gallà conceived and designed the experiments, performed the experiments, analyzed the data, performed the computation work, prepared figures and/or tables, authored or reviewed drafts of the paper, and approved the final draft.

The following information was supplied regarding data availability:

The virtual machine with the benchmark data and the virtual machine with the tools are available at Zenodo:

Elvio Gilberto Amparore, Susanna Donatelli, & Francesco Gallà. (2021). Replication Package (Virtual Machine) for article “starMC: an automata based CTL ∗ model checker” [Data set]. In PeerJ Computer Science (preprint). Zenodo. https://doi.org/10.5281/zenodo.5752419. The provided benchmark can be run either in quick or in full mode, which will take about 30 min/10 days, respectively, to complete.

The source code of the tool is available at GitHub: https://github.com/greatspn/SOURCES.

The benchmark data is available at GitHub: https://github.com/amparore/starMC-benchmark.

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
