# Peer review of "starMC: an automata based CTL* model checker"

_PeerJ Computer Science, doi:10.7717/peerj-cs.823_

## Round 0.1 · original submission · Major Revisions

Please carefully address the comments of all reviewers, particularly the comments concerning deadlock-freedom and fairness as well as presentation-related issues pointed out by the reviewers.

·

Basic reporting

This paper presents the ideas and algorithms behind the starMC tool, one of the few CTL* in existence. The tool uses decision diagrams for its symbolic implementation and accepts Petri nets as input. The implementation is evaluated in extensive experiments, using many models and formulas. This paper builds on an earlier demo paper, and extends it with separate logical and implementation views of the tool. Furthermore, it contains extensive references to original works, from which the ideas originated. The experimental evaluation is more rigorous than in the previous paper.

The paper is very clearly written and pleasant to read. The paper is mostly self-contained. There are plenty of references to previous work; the related work section covers more than two pages. I think this gives a lot of insight into the history of CTL* model checking and also its current state. The source code and files used for the experiments are available online, which is commendable.

Experimental design

The main questions posed in the paper are whether it's possible to design and implement a CTL* model checker and whether techniques like variable ordering and the extraction of counter-examples can be applied to it. As mentioned in the intro, the authors only answer the first half of this question, the second half is left for future work.

In my view, the setup of the experiments is appropriate for the question that the researchers try to answer. The set of benchmarks is based on the widely used MCC collection of models and formulas. The experimental results indicate that this is a varied set of benchmarks, which helps validity. My only criticism here is that the first research question relates to models of 'industrial interest', while it's not clear that such models are contained in the MCC benchmark set.

Validity of the findings

I think the main achievement of this paper is not the finding that CTL* model checking is possible in practice, but the fact that there is now a tool to do just that. The conclusion that starMC can be used for large state spaces is fair, given the experimental results. The authors also highlight the shortcomings of their tool.

Additional comments

The paper was truly enjoying to read, and I think that not much needs to be done for acceptance. I do have a few small remarks, which are listed below. The paper can be accepted when these are addressed.

Minor remarks/typos:
- p2, l92: distint -> distinct
- Sect. 2.2: in my opinion, it's nice to be consistent with singular/plural of abbreviated nouns. You use the abbreviation MDD both for singular and plural, but for plural of MxD, you use MxDs.
- p3: it might help the reader if you emphasize that the 'fully reduced rule' of skipping levels does not apply to MxDs. For example, the MxD of T0 does not restrict the relation between P2 and P2' if interpreted as fully reduced.
- Def. 7: braces for sets in the wrong place; should be Q = S \cup {Spre}, with Spre \notin S.
- p8, l287: s1 -> s_1
- Alg.3, l11: it was not immediately clear to me that this line is a comment. Perhaps there is a better syntax?
- p13, l363: Some aspects of this procedure needs -> Some aspects of this procedure need
- p13, l370: if M ⊗ A is a BA [..] weak or terminal -> if M ⊗ A is a weak or terminal BA
- p13, l374: "whenever Qi ≤ Qj , there is no transition from Q j to Q i." Perhaps this should be the other way around? Otherwise the partial order is allowed to be the identity relation.
- p13, l385: "onto the marking of s j." -> Do you mean "onto the marking of s_i"?
- Alg. 5, l9: perhaps you mean S = S'?
- p16, l528: "Most computations of the model checker works with DD" -> Most computations of the model checker work with DDs
- p17, l548: It is not immediately clear what "2n + 2 resp." refers to.
- Tables 1, 2 and 3: "Both terminates" -> Both terminate
- p23, l742: "LTSmin is faster then starMC" -> than

Reviewer 2 ·

Basic reporting

- The treatment of deadlocks is unclear. On one hand, a Kripke structure is defined (lines 169-170 and 173-175) to be deadlock-free, enforcing infinite paths. On the other hand, the experiments section reports "stuttering enabled for LTL, and disabled for CTL" (line 631-632), which seems inconsistent with the given definition for a Kripke structure, and begs the question of what then the decision is for CTL* and whether the decision for LTL and CTL follows that of line 631.

- Fairness is presented in a confusing manner. The acceptance sets of the constructed GBA are used as "fairness constraints" (line 14 of Algorithm 4), but given that I have limited knowledge of and experience with fairness assumptions, why this is the case is completely opaque to me. A clarification either way would likely help make the theory much more digestible to a reader not familiar with the original work by Emerson and Lei. The role of the fairness assumption could also be clarified to make the algorithm more transparent.

- Another avenue for improving transparency of the algorithm is a more thorough overview of the algorithm before going into the details. Lines 300-314 attempt this, but the takeaways from the example are not particularly clear, as lines 303-310 mainly demonstrate how the example cannot be checked just with LTL or CTL checking. More elaboration on how the different subformulae are dealt with would be nice here.

- The there are a handful of typos. Definition 3 lists that $s \models a$ iff $a \in L(a)$ when clearly it should be $a \in L(s)$, and Algorithm 5 writes $\mathbf{repeat} ... \mathbf{until} S \neq S'$ as a fixed-point computation, when it probably should be $\ldots \mathbf{until} S = S'$. The example on line 302 has a formula mentioning APs alpha and beta, but the text describe an AP gamma which is not present.

- Algorithm 5 details a function $CheckE_{fair}G$ that returns a sat-set, while other functions that compute sat-sets are called e.g. $SatCTL*$ (algorithms 2, 4).

Experimental design

- The comparison to LTSmin is not particularly convincing. The paper does not appear to give a convincing argument why it is sane to consider the state space generation step an important part of the verification procedure. Line 638/639 specifies 60 seconds for state space generation and 60 seconds per formula, but the paper focuses mainly on the formula verification half of this, in which case the state space generation step is not important. Because of this, the comparison to LTSmin should probably have been restricted to models where both starMC and LTSmin generated state spaces, rather than models where either of them did. The text also places some emphasis on the worst-case exponential translation to $\mu$ calculus done in LTSmin, it should be clarified whether this time was contained in the 60 seconds for state space generation or the 60 seconds per query.

- The method of constructing CTL* queries from MCC should be noted already on line 633, right now it's first introduced on line 735-738. The sentence on line 633 should also be revised, it seems to have grammatical issues.

- For sake of completeness the used Spot/Meddly versions and LTSmin version should be noted.

- It seems “unfair” that LTSmin gets 4 cores. Sylvan can utilize 4 cores, but it seems be an unfair comparison for starMC (which is single core) and comparing the underlying DD framework rather than the algorithm (translation via mu-calculus vs. the Emmerson-Lei Algorithm).

Validity of the findings

- The experimental setup in its is missing. The generated data is provided, but the experiment cannot be repeated w/o significant effort by a peer. This is particular problematic as LTSmin is forced to adopt a specific variable-ordering, which is known to be a significant factor in the efficiency of symbolic modelcheckers.

- As the author motivates their work by the rareness of existing CTL* modelcheckers for which executable code is available, the authors should ensure that their entire setup is provided in a repeatable format. This would include both source-code, models and scripts for execution and data-collection. Currently links are provided to the raw result (but not scripts, models or queries) along with links to mutable/volatile places on the internet. A link to a permanent data-storage service such as Zenodo is strongly recommended.

- "the experiments conducted in the testing phase indicate that starMC can
solve (significantly) more models than LTSmin" is a strong statement given the selection-criteria of the models. The followup comment on the speed of LTSmin also seems arbitrary, given the difference in resource allocation for the two tools.

Additional comments

The paper is recommended to be revised, in particular the experimental section.

Furthermore, the text as a whole could have been written more clearly and needs an additional proof-reading pass to make it clearer, a few examples are provided in the notes.

Reviewer 3 ·

Basic reporting

The paper is detailed and gives mostly details on the implementation of the starMC tool which is part of the greatSPN graphical petri net tool, although it can also be used standalone. The paper describes the tool and the implemented algorithms in sufficient detail. There is plenty of background and there is also a reasonable section on the related work in the literature. The work is self-contained. Not many proofs are included but this is not necessary since there are no true new algorithms proposed, merely existing algorithms that have been implemented using symbolic methods.

Virtual machine with the tool, csv summaries of the benchmarks etc can be downloaded, although it is a bit tricky because the link in the review form is broken.

Experimental design

Research questions are sufficient and the conclusions appear supported by the empirical evidence. They appear to be replicable. I would however recommend that the authors store an artifact of the benchmarks, raw data, processing scripts on an external platform such as Zenodo for long term persistence.

Validity of the findings

As reported in section 2, I think the conclusions are supported by the evidence, they appear to be scientifically accurate.

Additional comments

I found a minor spelling error "Sumamry" in the caption of Table 3. Other than that the paper is reasonable. I would personally have appreciated more discussion of variable orderings and their impact because I like that in BDD papers, but in this case I can see that it does not add much to the paper.

---

## Round 0.2 · accepted · Accept

Reviewer 1 is now happy with the revised manuscript. Reviewer 2 was unfortunately not available to check the revision; I, the handling editor, went through the comments and the revision and I am happy to recommend acceptance.

·

Basic reporting

This paper presents the ideas and algorithms behind the starMC tool, one of the few CTL* in existence. The tool uses decision diagrams for its symbolic implementation and accepts Petri nets as input. The implementation is evaluated in extensive experiments, using many models and formulas. This paper builds on an earlier demo paper, and extends it with separate logical and implementation views of the tool. Furthermore, it contains extensive references to original works, from which the ideas originated. The experimental evaluation is more rigorous than in the previous paper.

The paper is very clearly written and pleasant to read. The paper is mostly self-contained. There are plenty of references to previous work; the related work section covers more than two pages. I think this gives a lot of insight into the history of CTL* model checking and also its current state. The source code and files used for the experiments are available online, which is commendable.

Experimental design

The main questions posed in the paper are whether it's possible to design and implement a CTL* model checker and whether techniques like variable ordering and the extraction of counter-examples can be applied to it. As mentioned in the intro, the authors only answer the first half of this question, the second half is left for future work.

In my view, the setup of the experiments is appropriate for the question that the researchers try to answer. The set of benchmarks is based on the widely used MCC collection of models and formulas. The experimental results indicate that this is a varied set of benchmarks, which helps validity. My only criticism here is that the first research question relates to models of 'industrial interest', while it's not clear that such models are contained in the MCC benchmark set.

Validity of the findings

I think the main achievement of this paper is not the finding that CTL* model checking is possible in practice, but the fact that there is now a tool to do just that. The conclusion that starMC can be used for large state spaces is fair, given the experimental results. The authors also highlight the shortcomings of their tool.

The experimental results are easy to reproduce with the virtual machine the authors provided. The VM includes a script for running a subset of the experiments in a reasonable amount of time. The results appear to match those in the paper.

Additional comments

I want to thank the authors for carefully addressing all the comments I listed for the initial version.